# "Does Your Mobile Suit Your Skin?": Addressing Skin Tone Disparities in Presentation Attack Detection for Enhanced Inclusivity of Smartphone Security

## Abstract

Mobile devices are at a heightened risk for cybercrime due to the sensitive personal and financial data they handle. Biometric authentication provides a robust, convenient, and secure way to protect smartphones by using unique user characteristics like fingerprints, facial features, or voice patterns for access. Existing mobile biometric technology often relies on RGB cameras to capture biometric samples, such as face images or finger photos, making them vulnerable to spoofing (e.g., 3D masks, display, or printout attacks).

The security of these systems is effectively addressed by integrating a Presentation Attack Detection (PAD) module. Existing PAD solutions do not account for diverse physical characteristics like skin tone. As a result, marginalized groups face higher misidentification rates or false rejections, reducing access to services and increasing security risks.

This paper introduces a deep learning framework called ColorCubeNet designed to process *ColorCube*, a multi-dimensional data representation by combining information from RGB, HSV, and YCbCr color spaces. This data cube leverages the joint capabilities of RGB, HSV, and YCbCr color spaces to depict color more sophisticatedly. By incorporating features from multiple complementary color channels, this approach can effectively handle a variety of skin tones. We utilized three EfficientNet-B0 models, each trained on ImageNet using RGB, HSV, and YCbCr color spaces, and then fine-tuned them on the *ColorCube* representation to fully exploit the combined information from all three color spaces. Additionally, a channel-attention mechanism is integrated into the architecture, enabling the extraction of key features from different input channels by exploiting their combined performance. Results show that the proposed approach outperforms traditional RGB methods by reducing skin tone disparities by 50%.

## 1 Introduction

Through our smartphones, we handle and transmit personal data, including financial records, increasing the motivation for malicious individuals to launch attacks. As the use of these devices becomes more widespread, it is crucial to comprehend their vulnerabilities and enhance their defenses to uphold user trust and safeguard critical data Sta; Alrawili et al. (2024). Biometric-based unlocking mechanisms are susceptible to Presentation Attacks (PAs), where malicious actors attempt to circumvent security by presenting fake biometric samples, including photographs, masks, fake silicone fingerprints, and video replays that undermine the security of the system Ramachandra & Busch (2017); Marasco & Vurity (2022). Examples of bonafide and PA samples of face and finger photos are shown in Fig. 1. To mitigate these threats, it is common practice to integrate a Presentation Attack Detection (PAD) module into the biometric system to strengthen security Marasco & Ross (2014); Turhal et al. (2024); Purnapatra et al. (2023); Priesnitz et al. (2024). PAD is a vital component of mobile biometric authentication. Effective PAD technologies must be robust against various spoofing techniques, ensuring that only biometric samples acquired by living individuals physically present at the authentication point are accepted.

Mobile technology increasingly uses optical sensors, like RGB cameras, to capture biometric data for verification or identification via machine learning models Rattani & Derakhshani (2018); Ramoly et al. (2024); Li, Hailin and Ramachandra, Raghavendra and Ragab, Mohamed and Mondal, Soumik and Tan, Yong Kiam and Aung, Khin Mi Mi (2024). Technological discrimination arises when optical sensors fail to capture features accurately, especially for individuals with highly pigmented skin. This issue is prominent with RGB imaging and deep learning models used for biometric recognition, reducing accuracy and reliability for users with darker skin tones Linghu et al. (2024); Kinyanjui et al. (2019); Schlessinger (2023); Booysen & Theart (2024).

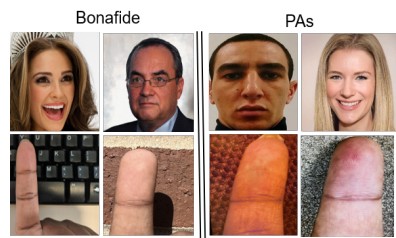

Figure 1: Bonafide and Presentation Attacks (PAs) samples.

These differentials compromise security and expose marginalized groups to greater risks of being unfairly denied access. Analyzing the impact of skin tone on these technologies is crucial for developing equitably secure mobile authentication systems Phelps (2021). Despite the critical role that PAD systems play in mobile security, there is currently no systematic assessment of how skin tone affects them. Furthermore, the existing PAD databases do not adequately provide skin tone data to facilitate this research. These limitations make it difficult to fully understand how different skin tones affect the accuracy and robustness of PAD systems, hindering the development of fair and effective solutions for all demographic groups.

Smartphone companies have responded to concerns about skin tone bias by launching new camera models with improved capabilities to capture and identify individuals with darker skin tones accurately Koenigsberger (2021); Meg (2023). Nonetheless, significant challenges remain for all optical sensors, particularly the less advanced ones. Technology must be developed considering people of all skin tones, and diverse teams must be involved throughout development. Whether and how PAD systems handle different skin tones can mitigate ethical and security concerns is understudied. To address this gap, the proposed research evaluates whether existing PAD technologies are equally effective for all users, regardless of skin tone. Furthermore, it investigates strategies to fine-tune the AI models to enhance their performance across diverse skin types. The objective is to address and rectify these vulnerabilities, ensuring that security technologies offer equitable protection to users from all backgrounds.

The proposed research aims to investigate the effectiveness of current PAD technologies for all users and their ability to recognize features across different skin tones accurately. This paper aims to assess and improve inclusivity in cybersecurity by examining how skin tone affects the accuracy and reliability of finger photo and face PAD systems. After assessing these disparities, we also explore how these technologies can be enhanced. We explore mitigation strategies that improve the inclusivity and accuracy of facial and finger photo technologies. This includes retraining AI models with diverse datasets that better represent all skin tones. Additionally, enhancing PAD techniques to be more effective across a broader range of conditions and skin types can help safeguard against PAs while ensuring fair treatment for all users. By focusing on these areas, we aim to create more reliable and equitable mobile biometric authentication systems that can be trusted in critical applications.

Previous studies have successfully integrated various color spaces, such as HSV, LAB, and YCbCr, to enhance Presentation Attack Detection (PAD) performance Marasco & Vurity (2022). Furthermore, a Person's based correlation analysis of these color spaces has demonstrated their complementarity (i.e., low correlation) Marasco & Vurity (2022). Each color space channel provides distinct information that can improve the robustness and accuracy of neural networks in managing color variations and generalizing across different inputs Lengyel & et al. (2023). Building on this promising direction, the proposed approach introduces a unified representation called *ColorCube*, combining nine channels (RGB, HSV, and YCbCr) to minimize the impact of skin tone variations. This representation captures data that is resistant to changes in skin tone.

This study introduces ColorCubeNet, which uses three EfficientNet-B0 models trained on different color spaces (RGB, HSV, and YCbCr) from scratch. These models are fine-tuned on the proposed *ColorCube* representation. We evaluated the performance of ColorCubeNet against traditional RGB-based models, demonstrating its effectiveness in reducing skin tone disparities. An extensive

evaluation is conducted using six different databases that cover face and finger photos and a variety of PAIs.

The contributions of this paper are summarized as follows:

- *Assessment of Skin Tone Impact on PAD*. We analyze skin tone differentials for facial and finger photo recognition systems in PAD. It is possible that the current PAD models are not effectively optimized for different skin tones, leading to unequal performance and heightened vulnerability. This can potentially cause security issues for specific groups. It is crucial to comprehend this impact to ensure consistent performance of PAD systems across diverse populations, thereby promoting fairness.

- *ColorCube Representation*. The proposed three-dimensional representation combines spatial information with nine color channels, including RGB, HSV, and YCbCr. This unified representation can detect subtle variations and features that traditional RGB-based systems miss, leading to improved and more accurate performance in PAD. The findings indicate that ColorCube features effectively reduce skin tone differentials.

- *Skin Tone Image Labeling*. Since PAD databases lack existing skin tone analysis, we labeled skin tones on finger photos and the facial database. This labeling process will categorize biometric samples by skin tone, enabling training PAD systems and ultimately improving generalization and fairness across diverse populations.

- *Training Paradigm-Shift*. We present ColorCubeNet, a new framework that retrains the backbone CNNs ( EfficientNet-B0) from scratch using ImageNet data converted into the *ColorCube* described earlier. ColorCubeNet also incorporates Channel Attention mechanisms, crucial for identifying the most significant features among the multiple color spaces. The channel attention mechanism extracts relevant channel-wise features that capture subtle differences in skin tone. This technique has been successfully used for skin disease detection in networks like Effi2Net Karthik et al. (2022).

- *Pioneering Explainable AI (XAI) to Interpret Skin Tone*. To the best of our knowledge, XAI techniques are combined with a signal-to-noise ratio (SNR) approach for the first time to analyze and identify the impact of skin tone on PAD systems. By applying XAI, we can interpret and understand the model's decision-making process, gaining valuable insights into how skin tone affects PAD performance. This approach not only aids in fine-tuning models for improved accuracy across different skin tones but also enhances transparency in the model's predictions, thereby increasing trust and fairness in the system.

## 2 LITERATURE REVIEW

### 2.1 IMPACT OF SKIN TONE IN BIOMETRICS

One of the most widely adopted skin tone measurements is the Fitzpatrick scale (FST), designed to assess UV sensitivity Sommers et al. (2019); Hazirbas et al. (2021a). In computer vision, apparent skin tone (AST) is commonly used to measure skin tone in images. The Individual Typology Angle (ITA) is a crucial metric for quantifying skin tone based on the CIE Lab* color space using L* (lightness) and B* (yellow/blue) values Krishnapriya et al. (2022). Additionally, the Monk Skin Tone (MST) scale, proposed by Monk, offers another widely recognized method for categorizing skin tones, which can provide further insights into the variability of skin tone representation Monk (2019). Recent studies have explored how biometric systems perform across various demographic groups Drozdowski et al. (2020), primarily focusing on covariants like gender, age, and ethnicity. However, there has been limited research on the specific impact of skin tone on face recognition Krishnapriya et al. (2022); Pangelinan et al. (2024). Krishnapriya *et al.* demonstrated that skin tone, categorized using the Fitzpatrick (1988) scale, influences the False Match Rate (FMR) in face recognition algorithms Krishnapriya et al. (2020).

More recent research has examined the role of gender, age, and ethnicity in PAD systems Yu et al. (2020); Karkkainen & Joo (2021); Fang et al. (2024); Ramachandra et al. (2022); Trinh & Liu (2021); Xu et al. (2022); Nadimpalli & Rattani (2022); Hazirbas et al. (2021b); Ju et al. (2024); Kotwal & Marcel (2024). Still, the impact of skin tone, distinct from ethnicity, remains underexplored.

Our research aims to fill this gap by investigating how skin tone affects PAD systems, recognizing that skin tone can vary within ethnic groups and impact the detection of PAs.

Fang *et al.* (2024) introduced the Combined Attribute Annotated PAD Dataset (CAAD-PAD) to evaluate fairness in face PAD systems Fang et al. (2024), highlighting the need for fairness-aware models. Their findings revealed that certain demographic groups, like females and individuals with occluding features, are less protected by existing PAD solutions. While fairness studies have focused mainly on face recognition algorithms, the fairness of PAD systems has been largely overlooked, with limited attention given to this aspect in face PAD systems. In recent studies, we see a growing trend for balancing fairness, interpretability, and privacy in AI systems by emphasizing the importance of XAI to ensure ethical, transparent, and bias-free models, especially in sensitive domains Longo et al. (2024); Zhou et al. (2020); Ferry et al. (2024)

## 2.2 LITERATURE ON FACE AND FINGER PHOTO PAD

*Finger photo PAD:* Finger photo presentation attack detection (PAD) systems have evolved from traditional methods using handcrafted features such as texture patterns and gradients, often classified with machine learning algorithms like Support Vector Machines (SVMs) Guo et al. (2010); Kannala & Rahtu (2012); Lowe (1999); Hearst et al. (1998). These approaches, however, struggled to generalize across different devices and attack scenarios. Deep learning models, particularly CNNs, have recently become more prominent due to their superior performance. Marasco *et al.* proposed a framework that segments the finger region, converts it into multiple color spaces, and analyzes local patches around minutiae points using an ensemble of pre-trained CNNs Marasco & Vurity (2022), significantly improving PAD robustness against spoofing attacks. Li *et al.* Li & Ramachandra (2023) compared various deep learning architectures, such as DenseNet, ResNet, and EfficientNet, highlighting the advantages of deep learning in improving detection accuracy. Additionally, Adami *et al.* developed an unsupervised finger photo PAD method using an autoencoder and convolutional block attention, achieving a BPCER of 0.96% and an APCER of 1.6% Adami & Karimian (2023). In 2024, researchers evaluated eight pre-trained deep neural network models across different finger segmentation schemes on a public dataset featuring four presentation attack instruments Li & Ramachandra (2024).

*Face PAD:* Face PAD has become essential in biometric security systems due to increasing spoofing attacks. Early methods relied on handcrafted features like Local Binary Patterns (LBP), Histogram of Oriented Gradients (HOG), and color texture analysis, combined with classifiers such as SVMs Song & Liu (2018); Pereira et al. (2012); Chingovska et al. (2012); Maatta et al. (2011). While effective in controlled settings, these methods struggled with varying conditions and attack types. Deep learning, particularly CNNs, has provided more robust solutions by learning complex features directly from raw data Yu et al. (2023); Maphisa & Coulter (2022); Xu et al. (2017); George & Marcel (2019); Atoum et al. (2017); Zhao et al. (2017). Koshy *et al.* demonstrated the effectiveness of ResNet-50 and Inception v4 for face PAD, improving spoof detection across datasets Koshy & Mahmood (2019). Xu *et al.* showed that combining CNNs with LSTM enhances facial anti-spoofing in videos Xu et al. (2015). Additionally, transformers have been employed to explore bonafide-PA relationships among local face patches in the spatial domain et al. (2021); Wang et al. (2021); Chen et al. (2022) and extract global features related to temporal abnormalities in the temporal domain Liu & Pan (2024).

## 3 OVERVIEW OF THE PROPOSED FRAMEWORK

Conventional approaches often fail to capture the subtle chromatic variations in skin pigmentation across diverse skin types, leading to inaccuracies. To address this limitation, we propose *ColorCubeNet*, a framework that integrates information from multiple color spaces, specifically RGB, HSV, and YCbCr called ColorCube. By processing nine-channel images derived from these color spaces, the model leverages the EfficientNet-B0 backbone to extract robust and hierarchical features. These features are further refined using a channel attention mechanism, which dynamically emphasizes the most relevant feature channels. Each color space contributes uniquely to the analysis by highlighting different aspects of color information, enhancing the system's ability to handle a wide range of skin tones Prema & Manimegalai (2012). In this research, we are chosing RGB, HSV and

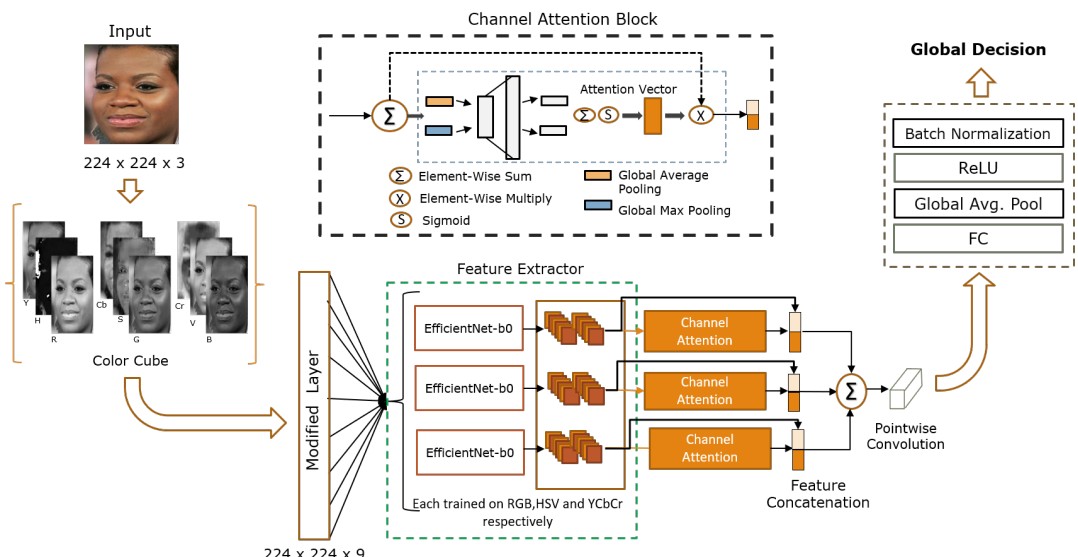

Figure 2: ColorCubeNet Architecture

YCbCr in the analysis as previous PAD papers have proven these three colorspaces have greater impact Marasco & Vurity (2022; 2021).

The EfficientNet-B0 backbone is a natural choice for this framework because of its efficiency and scalability. Designed with a compound scaling method, EfficientNet-B0 achieves high performance with minimal computational overhead, making it particularly suitable for resource-constrained environments Tan & Le (2019). The channel attention mechanism plays a crucial role in refining the extracted features. By dynamically weighting the feature channels, it allows the network to focus on the most informative aspects of the input, which is especially important when processing nine-channel images. This selective focus helps to enhance the subtle differences in chromatic information that are critical for distinguishing between bona fide and presentation attacks.

An overview of the proposed architecture is illustrated in Fig. 2. The model converts an input face or finger photo RGB image of size 224×224×3 into a *ColorCube* size 224x224x9. The framework is based on three parallel EfficientNet-B0 with channel attention. The outputs from the individual channel-attention blocks are concatenated to obtain the features that are then processed through the final layers to make predictions (bonafide or PA). The details of the computational complexity is provided in Appendix A5.

## 3.1 COLORCUBE DERIVATION

This section discusses the mathematical derivation of the proposed *ColorCube*. Let capture device $C$ be defined by $C : T \rightarrow P$, where $T$ denotes triggering the smartphone camera and $P$ indicates the face or finger modality being captured. $P(x, y) = [R(x, y), G(x, y), B(x, y)]$ represents the resulting image pixels in the RGB color space, where $(x, y)$ are coordinates of the pixels. Let $I(x, y) = [R(x, y), G(x, y), B(x, y)]$ be the image captured by the sensor $S : T \rightarrow I$.

*ColorCube Representation*: The RGB pixel values at coordinates $(x, y)$ are transformed into two additional color spaces, HSV[H(x,y), S(x,y) and V(x,y)] and YCbCr[Y(x,y), Cb(x,y), Cr(x,y)] to create a unified *ColorCube* representation. This results in a 9-channel vector for each pixel:

$$
\begin{aligned}
C(x, y) = [&R(x, y), G(x, y), B(x, y), \\
&H(x, y), S(x, y), V(x, y), \\
&Y(x, y), C_b(x, y), C_r(x, y)]
\end{aligned}
\tag{1}
$$

This ColorCube combines the strengths of all three color spaces, enabling a richer representation of color variations across different skin tones. The final representation is normalized and converted into an input tensor for the model. Detailed color space math is shown in Appendix A2.

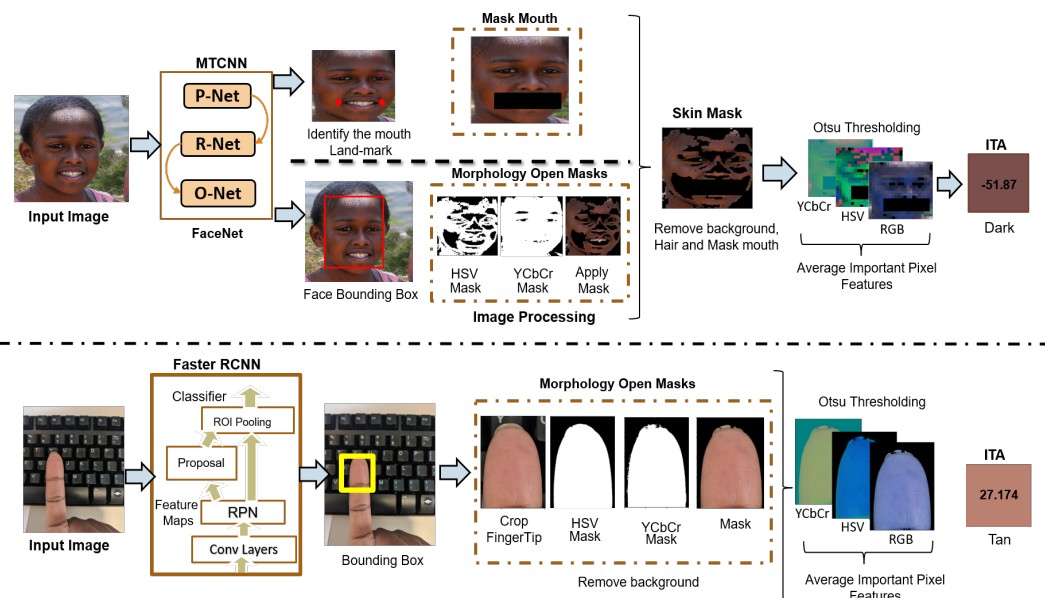

Figure 3: Face and Finger Photo Skin Tone Detection Pipeline

## 3.2 SKIN TONE IMAGE LABELING

Skin tone classification is crucial in understanding the impact of skin color on PAD systems. One of the most widely used methods for quantifying skin tone is the Individual Typology Angle (ITA), derived from the Lab* color space Chardon et al. (1991). The ITA provides a continuous, objective means of categorizing skin tones based on their lightness ($L^*$) and blue-yellow chromaticity ($b^*$) components. The ITA, expressed in degrees, is calculated using the following equation:

$$\text{ITA} = \arctan\left(\frac{L^* - 50}{b^*}\right) \times \frac{180}{\pi} \tag{3}$$

ITA values are then categorized into predefined ranges, known as the Apparent Skin Tone (AST) scaleKrishnapriya et al. (2022), which provides the basis for assessing model performance across various skin tones (see Table 2 in Appendix (A1). The skin tones include Brown (B), Dark (D), Intermediate (I), Tan (T), Light (L), and Very Light (VL).

We employed a dual-process approach for detecting skin tone in both face and finger photos, as shown in Fig 3. For facial skin tone detection, we used the FaceNet architecture with Multi-task Cascaded Convolutional Networks (MTCNN), including Proposal (P-Net), Regional (R-Net), and Output (O-Net) networks, to detect faces and locate facial landmarks Schroff et al. (2015). After identifying the face, a mask is applied to the mouth region to avoid interference with skin tone estimation. Morphological operations are then applied in HSV and YCbCr color spaces to generate masks. Otsu thresholding is used to separate skin from non-skin regions, and the average pixel values from HSV and YCbCr are used to extract essential skin tone features. From these features, the Individual Typology Angle (ITA) is computed.

For fingertip localization, we used a faster R-CNN model fine-tuned on finger photos Marasco & Vurity (2021). This model includes a Region Proposal Network (RPN) and Region of Interest (ROI) pooling to generate a bounding box around the fingertip. Like the facial detection pipeline, background removal is performed using morphological operations in HSV and YCbCr color spaces. Otsu thresholding is again applied to refine the skin mask, and the extracted pixel values are used to compute the ITA, allowing us to classify each sample into the Apparent Skin Tone (AST) categories.

## 3.3 CONVOLUTIONAL BACKBONE AND CHANNEL-ATTENTION

ColorCubeNet's first layer is modified to accommodate *ColorCube*'s new input dimension, as illustrated in Fig. 2. Three EfficientNet B0 models are used as the backbone for feature extraction. Each

EfficientNet-B0 is trained from scratch on the ImageNet Dataset Deng et al. (2009) using RGB, HSV, and YCbCr color spaces. For each model $\Phi_i$, the feature map $\Phi_i(C_{\text{tensor}}(x, y))$ is computed as:

$$\Phi_i(C_{\text{tensor}}(x, y)) = \text{EfficientNet-B0}_i(C_{\text{tensor}}(x, y)),$$
$$i \in \{\text{RGB}, \text{HSV}, \text{YCbCr}\} \tag{4}$$

Feature maps $\Phi_i(C_{\text{tensor}}(x, y))$ extracted from the EfficientNet-B0 models are individually processed through a single channel-attention block $A$ to emphasize the most relevant features separately across different RGB, HSV, and YCbCr color space models.

$$A(\Phi_i(C_{\text{tensor}}(x, y))) = \sigma\left((\text{GlobalAvgPool}(\Phi_i(C_{\text{tensor}}(x, y)))\right.$$
$$+ \text{GlobalMaxPool}(\Phi_i(C_{\text{tensor}}(x, y)))) \cdot W + b)$$
$$\odot \Phi_i(C_{\text{tensor}}(x, y)) \tag{5}$$

Here, $\sigma$ is the sigmoid function, scaling values between 0 and 1. Learnable parameters $W$ and $b$ adjust during training, while element-wise multiplication $\odot$ applies the attention weights to the feature map $\Phi_i(C_{\text{tensor}}(x, y))$, prioritizing critical features across RGB, HSV, and YCbCr.

*Feature Concatenation*: As shown in Fig.2, the output feature maps from each attention block concatenated into a single combined feature map $F_{\text{concat}}$ the outputs from each channel attention block are processed individually and then combined using element-wise summation.

$$F_{\text{concat}} = A(\Phi(C_{\text{tensor}}(x, y))) \tag{6}$$

Subsequently, the refined features are passed through a series of operations, including batch normalization, ReLU activation, and global average pooling. Finally, a fully connected layer makes the final decision on whether to bonafide or PAs. To further analyze the model's decision-making, we applied Grad-CAM to visualize important input regions, and used Signal-to-Noise Ratio (SNR) to quantify the clarity of these key features across different skin tones. In this work, we use Grad-CAM Selvaraju et al. (2017) and SNR to isolate relevant features influencing PAD decisions (signal) from background noise, enhancing interpretability and reliability. This method can also be extended to other saliency map techniques. Quantifying these visualizations enables us to evaluate PAD performance across different skin tones.

The SNR quantifies the clarity of the signal in saliency maps, where higher SNR values indicate a more interpretable signal (greater than 1). This metric enables us to quantify the interpretability of PAD decisions and compare performance across different skin tones. It is calculated using the formula:

$$\text{SNR} = \frac{1}{N} \sum_{i=1}^{N} f_i \left/ \sqrt{\frac{1}{M} \sum_{j=1}^{M} (f_j - \mu)^2} \right. \tag{7}$$

where $f_i$ and $f_j$ represent the pixel intensities in the key activation and less relevant regions, respectively. $N$ and $M$ denote the number of pixels in these regions, and $\mu$ is the mean intensity of the less relevant region. This metric highlights the most important image regions used in PAD decisions and evaluates the model's performance across different skin tones.

## 4 Experiments and Discussions

To validate the effectiveness of our proposed ColorCubeNet architecture, we conduct experiments across six different datasets, each representing either face or finger photo biometric modality and presentation attacks.

### 4.1 Datasets

*CelebA-Spoof:* This dataset has over 625,537 pictures and 10,177 subjects, showcasing various spoofing attacks, including printouts, replayed videos, and 3D masks. To enhance the dataset's diversity, data was collected from five different angles and four distinct shapes using 24 popular devices, including PCs, cameras, tablets, and phones, with resolutions ranging from 12 to 40 megapixels Zhang et al. (2020).

OULU-NPU: The OULU-NPU dataset is a widely used benchmark of PAD. It consists of data from 55 subjects and includes 5,950 video clips captured using mobile devices. It contains both bonafide

and PAs, including printouts and video attacks. For our experiments, we used four frames per video that were extracted at specific intervals (10th, 30th, 80th, 140th) Boulkenafet et al. (2017).

*SynthASpoof:* The dataset supports face anti-spoofing research by providing computer-generated spoofing samples. This dataset consists of 25,000 bonafide subjects. There are 3,800 printout attacks and 75,000 images of replay attacks using a Webcam, Samsung phone, and iPad Fang et al. (2023).

*IIITD Smartphone Finger photo:* The dataset includes 64 subjects and 12,288 images, encompassing bonafide and PAs (spoofs) samples captured using smartphone cameras. The dataset features finger photos taken under various lighting conditions and backgrounds, with spoofing methods such as printout and display attacks using different devices Taneja et al. (2016).

*Finger Photo Presentation Attack Detection iPhone 13 Pro 2022 (FPAD-i-22)*: This dataset consists of 14,336 images in total, collected from 112 subjects. It has both bonafide and PA samples, with 2,688 Bonafide images and 11,648 PAs. The bonafide images were captured under various conditions, including indoor and outdoor environments, with variations in lighting and backgrounds (natural and white). The spoof samples were generated using display devices such as the Samsung Tab 7+, iPad Pro, and MacBook Pro and printout attacks using an HP Color-LaserJet MFP printer Vurity & Marasco (2023).

*Finger Photo Presentation Attack Detection Google pixel 3 2023 (FPAD-g-23)*: This dataset comprises 25,559 images in total, collected from 100 subjects. It includes bonafide and PAs with 4,000 bonafide images and 21,559 PA samples. The bonafide images were captured across various conditions, covering indoor and outdoor settings, with different lighting conditions and background variations Vurity & Marasco (2023).

## 4.2 EVALUATION PROTOCOL

To effectively adapt the baseline models to the PAD task, we utilized models pre-trained on the ImageNet dataset. These baseline models are fine-tuned using transfer learning by freezing the parameters and adjusting them to two classes (bonafide and PAs). The training protocol involved a batch size of 32, running for 30 epochs, with early stopping triggered after five consecutive epochs of no improvement. We applied data augmentation techniques such as horizontal flipping to enhance the training process further. Additionally, the datasets used are mutually exclusive subject-wise.

*Performance Metrics*: Attack Presentation Classification Error Rate (APCER), Bona Fide Presentation Classification Error Rate (BPCER) as defined by the International Organization for Standardization (ISO/IEC SC 37), and the Equal Error Rate (EER). APCER assesses the proportion of attack attempts mistakenly classified as Bona Fide, while BPCER measures the proportion of Bona Fide attempts incorrectly identified as attacks. In our results, we present the BPCER% when APCER% is set at 5% and 10%, respectively, to provide a detailed analysis of the model's performance. The receiver operating characteristic (ROC) curve allows us to visualize the trade-off between APCER and 1-BPCER at varying classification thresholds.

## 4.3 RESULTS

Fig 3 was applied to compute the ITA values for each dataset. We then proceeded with Principal Component Analysis (PCA) to reduce the dimensionality of the feature space. Here, the PCA is applied on the colorcube features. This approach allowed us to visualize the variability across different skin tones and analyze how the features extracted by Color-CubeNet are distributed along the principal components. Fig. 4 compares PCA results for RGB and ColorCube features applied to the FPAD-i22. The remaining PCA results are shown in Appendix (A4) Fig.6. In the RGB domain, the first prin-

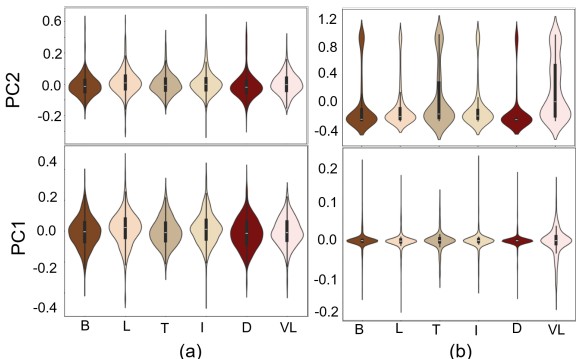

Figure 4: PCA on FPAD-i-22. (a) RGB (b) Colorcube

cipal component (PC1) shows high variability for Brown and Dark skin tones, while the *Color-Cube* features exhibit reduced variability, leading to more focused and consistent distributions. This pattern holds across the dataset. In the second principal component (PC2), RGB features display significant variability across skin tones, particularly for Dark tones. At the same time, the *Color-Cube* representation reduces this variability by centralizing the distributions and minimizing noise. This comparison demonstrates that the *ColorCube* representation offers a more stable and reduced-diversity feature space, making it the optimal choice for handling diverse skin tones in our proposed architecture.

### 4.3.1 EFFECTIVENESS OF COLORCUBENET AS PAD

Table 1: Performance comparison across different datasets at 5% and 10% BPCER. Baselines are ResNet 18 Marasco & Vurity (2021), ResNet 34Marasco & Vurity (2021), Resnet 101 Abdullakutty et al. (2022); Raja et al. (2023), EfficientNet-B0,B5,B7 Li & Ramachandra (2023), VIT-B-Patch16 Raja et al. (2023), DeiT Raja et al. (2023),George & Marcel (2019)

| Baseline | Celeb-A | | OULU NPU | | Synth-A-Spoof | | FPAD-i-2022 | | FPAD-g-2023 | | IIIT-D | |
|---|---|---|---|---|---|---|---|---|---|---|---|---|
| | 5% | 10% | 5% | 10% | 5% | 10% | 5% | 10% | 5% | 10% | 5% | 10% |
| ResNet 18 | 6.98 | 3.75 | 22.08 | 12.98 | 1.13 | 0.36 | 3.11 | 1.77 | 12.48 | 5.87 | 1.39 | 0.51 |
| ResNet 34 | 6.49 | 3.91 | 19.78 | 10.90 | 0.57 | 0.17 | 1.45 | 0.61 | 8.51 | 3.77 | 1.27 | 0.56 |
| ResNet 101 | 9.13 | 4.83 | 20.00 | 10.98 | 0.31 | 0.08 | 3.20 | 1.45 | 7.26 | 2.43 | 1.46 | 0.39 |
| EfficientNet-B0 | 49.36 | 35.16 | 23.25 | 11.83 | 0.11 | **0.03** | 3.03 | 1.28 | 15.12 | 7.72 | 1.54 | 0.51 |
| EfficientNet-B5 | 61.52 | 47.35 | 33.88 | 21.27 | 19.64 | 11.10 | 4.69 | 2.26 | 13.80 | 8.05 | 1.11 | 0.44 |
| EfficientNet-B7 | 51.76 | 38.98 | 26.54 | 14.99 | 13.18 | 6.95 | 5.76 | 3.46 | 16.83 | 7.07 | 5.83 | 2.64 |
| VIT-B-Patch16 | 79.03 | 64.99 | 39.03 | 29.51 | 0.87 | 0.32 | 29.69 | 17.72 | 8.88 | 5.25 | 34.99 | 16.46 |
| DeIT | 43.13 | 29.15 | 15.82 | 8.82 | 0.06 | 0.02 | 11.18 | 3.67 | 13.89 | 7.88 | 18.70 | 5.30 |
| DeePixBiS | 11.01 | 5.09 | 2.83 | 1.25 | 0.1 | 0.04 | 0.94 | 0.47 | 4.88 | 1.51 | 0.17 | 0.05 |
| **ColorCubeNet (Our)** | **3.28** | 1.19 | 1.88 | **0.50** | **0** | **0** | **0.34** | **0.13** | **4.75** | **1.38** | **0.05** | **0.02** |

Table 1 provides a comprehensive performance comparison of different baseline models across various datasets at 5% and 10% APCER. Our proposed ColorCubeNet model consistently outperforms the baseline models, achieving notably low BPCER values across most datasets. Specifically, on the Synth-A-Spoof dataset, ColorCubeNet achieves a BPCER of just 0.0% at 5% APCER and 0% at 10% APCER. The model also performs exceptionally well on FPAD-i-2022, Celeb-A, and IIIT-D datasets, maintaining BPCER values as low as 0.34%, 3.28%, and 0.05% respectively.

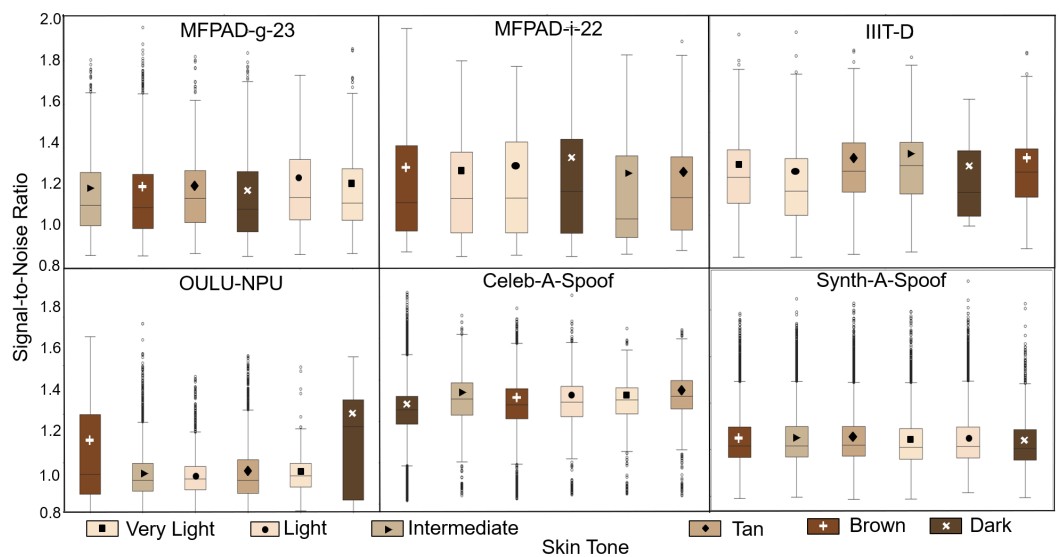

Figure 5: Signal to Noise Ratio (SNR) of ColorCubeNet on different datasets

In contrast, other models like EfficientNet-B5 and VIT-B-Patch16 show considerable variability across datasets, with EfficientNet-B5 recording higher BPCER on OULU-NPU (33.88% at 5% APCER) and VIT-B-Patch16 struggling with high BPCER values across most datasets (e.g., 39.03%

at 5% APCER on OULU-NPU). ResNet models and DeePixBiS also demonstrate mixed performance, with ResNet 34 performing relatively well on the Synth-A-Spoof dataset (0.57% at 5% APCER) but underperforming on OULU-NPU. DeePixBiS, though strong on Synth-A-Spoof, also shows higher BPCER values on other datasets. Overall, ColorCubeNet is consistent and has superior performance across all the datasets. To understand skin tone's impact on PAD systems, we analyzed mismatch rate and SNR on all six datasets. Our analysis clearly shows the effects of skin tone on PAD performance.

*Mismatch Rates Across Skin Tones*: This analysis highlights how skin tone disparities manifest in PAD systems. Fig. 7 in Appendix (A4) shows that traditional RGB models are more biased with darker skin tones. Fig. 8 demonstrates a significant reduction in mismatch rates (more than 50%) across all skin tones when using ColorCubeNet compared to traditional RGB models. This shows the effectiveness of the ColorCube representation in enhancing model generalization while maintaining inclusiveness across diverse skin tones. We focus on SNR scores derived from saliency maps using Grad-CAM to support this point further.

*Signal-to-Noise Ratio (SNR) Analysis:* In addition to mismatch rates, we evaluated SNR that quantitatively measures how well the model distinguishes between bonafide and attack features for different skin tones. Fig. 5 shows the distribution of SNR values for six datasets using ColorCubeNet. SNR values above 1 indicate that the model effectively captures distinct features, while values below 1 suggest difficulty in feature identification. The SNR results show that ColorCubeNet maintains balanced performance across various skin tones. For lighter tones (Light, Very Light), the SNR values remain consistently close to 1, indicating reliable performance in feature extraction. However, the analysis also reveals a slight dominance toward darker skin tones. In datasets like OULU-NPU and FPAD-g-23, the Brown and Dark skin tones exhibit slightly higher SNR values, suggesting that the model can effectively capture critical features for these tones. This dominance, while subtle, indicates that ColorCubeNet not only reduces disparities for lighter tones but also offers enhanced feature separation for darker tones. Thus, ColorCubeNet achieves a balance across the skin tone spectrum, which was previously underserved by traditional PAD models. While some variability still exists, particularly for very light and intermediate tones in specific datasets, the model demonstrates a clear improvement in reducing bias and enhancing fairness across skin tones.

*Ablation study*: Table 4 in Appendix (A5) presents the results of an ablation study on FPAD-g-23, evaluating backbone feature extractor trained on RGB, HSV, and YCbCr color spaces, channel-attention mechanisms, and feature concatenation blocks. The study shows that the ColorCubeNet model utilizes all the blocks to achieve the best performance with an accuracy of 97.34% and the lowest EER of 4.41%. Models using a single backbone or two backbones in parallel (e.g., RGB or RGB+HSV) or not utilizing feature concatenation or Channel Attention had higher EERs, ranging from 5.26% to 5.63%, indicating the benefits of incorporating all three backbones and attention mechanisms. Fig.9 illustrates various 1-BPCER values across various thresholds of APCER.

## 5 CONCLUSIONS

This paper highlights the importance of designing inclusive mobile PAD systems by addressing skin tone disparities. The proposed solution combines multiple color spaces to provide a more comprehensive skin tone representation. Results show that our framework consistently outperforms traditional RGB baseline approaches across six datasets—Celeb-A-Spoof, OULU-NPU, Synth-A-Spoof, FPAD-i-22, and FPAD-g-23 with BPCERs of 3.28%, 1.88%, 0%, 0.34%, 4.75%, and 0.05%, at 5% APCER, respectively. This performance shows that ColorCubeNet mitigates skin tone disparities better than traditional models.

Further analysis using Signal-to-Noise Ratio (SNR) confirmed the robustness of our model. SNR analysis showed stable values for lighter skin tones and improved feature separation for darker skin tones indicates a better handling of underrepresented skin tones. Despite these improvements, variability in SNR for very light and intermediate tones suggests, some challenges remain in achieving uniform performance across all skin tones.

We plan to explore additional color spaces to improve feature capture across all skin tones. We also aim to investigate further how variations in lighting conditions and capture devices influence the appearance of skin tones, which can significantly impact model performance. Additionally, we intend to incorporate other skin tone classification methods, such as the Monk and Fitzpatrick scales, to complement the Apparent Skin Tone (AST) method used in this study.

## 6 ETHICAL STATEMENT

Our study uses publicly available datasets with biometric data, and where applicable, we have ensured compliance with all necessary data privacy and ethical standards. Datasets used in this study may require prior approval or permission from the respective data owners for access. In such cases, we have signed license agreements or obtained Institutional Review Board (IRB) approval to ensure data privacy and compliance with ethical standards. No sensitive or personally identifiable information is publicly shared in this research, and we only provide irreversible extracted features that preserve privacy.

This study aims to mitigate biases in AI systems by leveraging the ColorCubeNet architecture, designed to improve PAD performance across various skin tones. No conflicts of interest or external sponsorships have influenced the results of this work, and all experiments were conducted in accordance with the highest standards of research integrity.

## 7 REPRODUCIBILITY STATEMENT

The datasets used in this study contain biometric images, which are sensitive in nature. To uphold data privacy and ethical guidelines, we are only sharing the code for our framework. Researchers interested in training the code on the original datasets must request access from the respective dataset authors, as cited in this paper.

We provide the full implementation of the ColorCubeNet architecture, the SkinTone pipeline, and the PCA analysis in the supplementary material to ensure complete transparency of the methods and processes. While the code for Signal-to-Noise Ratio (SNR) analysis is included, it requires subject-specific IDs and skin tone information, which may compromise privacy. To address this, we are offering a sample SNR code to illustrate its functionality without risking data exposure.

In addition, we have provided thorough documentation to guide users through reproducing our work. We are also sharing the models trained on the respective datasets, which can be evaluated by those who have been granted access to the datasets.

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

# A APPENDIX

## A.1 APPARENT SKIN TONE

Table 2: Skin Tone Classification based on ITA values within the AST method.

| ITA Range (°) | Skin Tone Classification |
|---|---|
| >55 | Very Light |
| 41 to 55 | Light |
| 28 to 41 | Intermediate |
| 10 to 28 | Tan |
| -30 to 10 | Brown |
| <-30 | Dark |

## A.2 *ColorSpace* DERIVATION

The transformation of the image from its original RGB color space into a combined RGB, HSV, and YCbCr representation begins with the derivation of the individual components. The **V** channel in the HSV color space is computed as the maximum of the RGB values at each pixel $(x, y)$:

$$V(x,y) = \max(R(x,y), G(x,y), B(x,y)) \tag{1}$$

The **S** (saturation) channel depends on the value of $V(x, y)$:

$$S(x,y) = \begin{cases} 0 & \text{if } V(x,y) = 0 \\ \frac{\Delta(x,y)}{V(x,y)} & \text{otherwise} \end{cases} \tag{2}$$

where $\Delta(x,y) = V(x,y) - \min(R(x,y), G(x,y), B(x,y))$. The **H** (hue) channel is calculated using the angle between the RGB components, and depends on the saturation $S(x, y)$:

$$H(x,y) = \begin{cases} 0 & \text{if } S(x,y) = 0 \\ \frac{\text{Angle}(R,G,B)}{\Delta(x,y)} & \text{otherwise} \end{cases} \tag{3}$$

For the YCbCr color space conversion, the RGB values are transformed into **Y** (luminance), **Cb** (blue chrominance), and **Cr** (red chrominance) components:

$$Y(x,y) = 0.299R(x,y) + 0.587G(x,y) + 0.114B(x,y) \tag{4}$$

$$Cb(x,y) = \frac{B(x,y) - Y(x,y)}{2} + 0.5 \tag{5}$$

$$Cr(x,y) = \frac{R(x,y) - Y(x,y)}{2} + 0.5 \tag{6}$$

These RGB, HSV, and YCbCr components are concatenated into a 9-dimensional vector for each pixel, which is normalized as follows:

$$\begin{aligned} C_{\text{norm}}(x,y) = \frac{1}{255} \times [&R(x,y), G(x,y), B(x,y), \\ &H(x,y), S(x,y), V(x,y), \\ &Y(x,y), Cb(x,y), Cr(x,y)] \end{aligned} \tag{7}$$

Finally, the normalized **ColorCube** representation is transformed into a tensor that is compatible with neural network input:

$$C_{\text{tensor}}(x,y) = \text{permute}(C_{\text{norm}}(x,y), \text{order} = [2, 0, 1]) \tag{8}$$

## A.3 CROSS DATABASE ANALYSIS

| Baseline | Train | Celeb-A | | OULU NPU | | Synth-Spoof | | Train | FPAD-i-22 | | FPAD-g-23 | | IIIT-D | |
|---|---|---|---|---|---|---|---|---|---|---|---|---|---|---|
| | | EER% | HTER% | EER% | HTER% | EER% | HTER% | | EER% | HTER% | EER% | HTER% | EER% | HTER% |
| Our | Celeb-A | 4.24 | 9.14 | 26.24 | 33.72 | 36.24 | 41.19 | FPAD-i-22 | 2.94 | 6.62 | 18.36 | 27.03 | 49.02 | 55.41 |
| | OULU NPU | 30.26 | 38.29 | 3.44 | 11.01 | 33.58 | 37.48 | FPAD-g-23 | 11.31 | 18.9 | 6.6 | 11.2 | 44.51 | 49.49 |
| | Synth-Spoof | 47.06 | 52.15 | 34.025 | 38.82 | 0 | 0 | IIIT-D | 36.79 | 43.61 | 27.01 | 35.89 | 0.95 | 2.77 |
| Deepixbis | Celeb-A | 18.21 | 27.17 | 31.76 | 38.45 | 47.61 | 52.29 | FPAD-i-22 | 7.11 | 18.79 | 31.54 | 38.49 | 56.25 | 60.02 |
| | OULU NPU | 40.16 | 48.47 | 27.59 | 35.21 | 39.59 | 45.65 | FPAD-g-23 | 26.11 | 32.47 | 8.78 | 17.48 | 35.63 | 43.37 |
| | Synth-Spoof | 53.24 | 64.51 | 46.25 | 50.01 | 0.95 | 13.07 | IIIT-D | 47.52 | 49.57 | 32.75 | 39.63 | 7.25 | 21.89 |

Table 3: Results show cross dataset analysis for both fingerphoto and face datasets.

Table 3 illustrates the cross-scenario evaluation, where a model trained on one face dataset is tested on other datasets. The results are reported in terms of EER% (Equal Error Rate) and HTER% (Half Total Error Rate). We compare the performance of ColorCubeNet against the best-performing baseline model.

## A.4 SkinTone Analysis

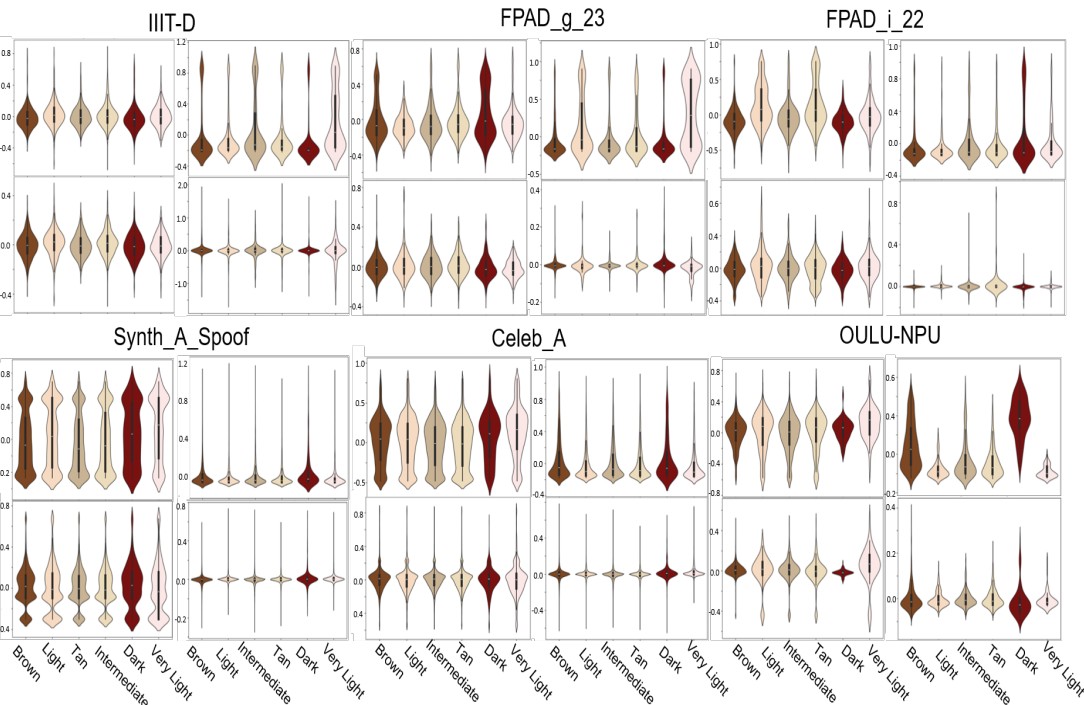

Figure 6: Distribution plots show Colorcube mitigating the skin tone impact.

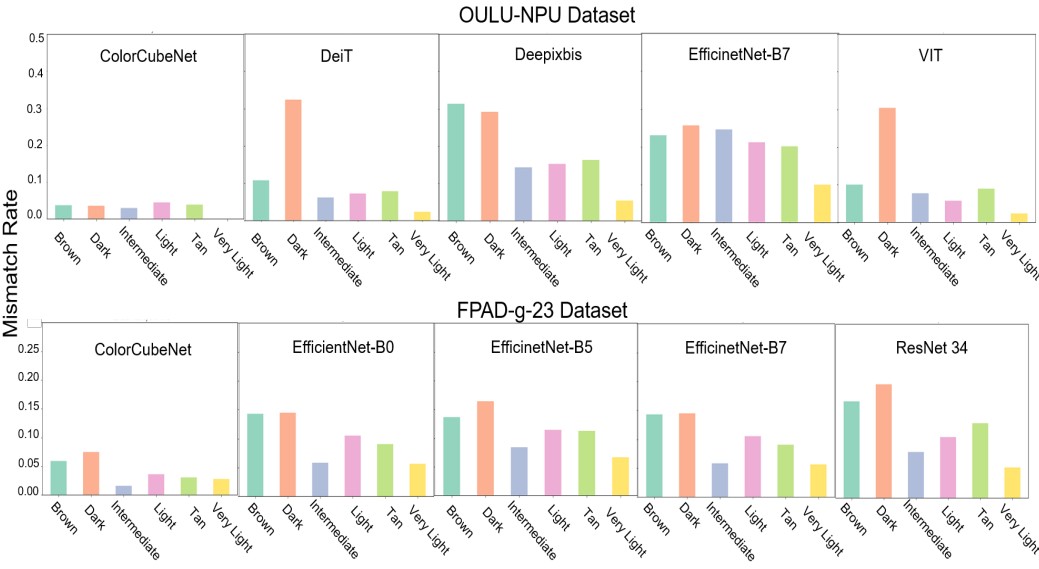

Figure 7: Mismatch rate of skin tones on OULU- NPU(Face) and FPAD-g-23 (Finger) datasets.

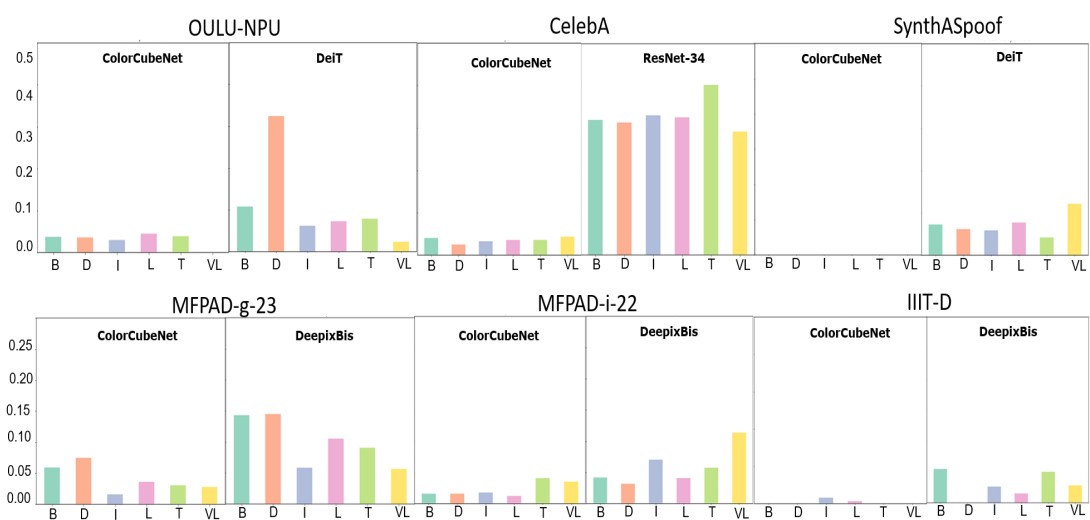

Figure 8: Mismatch rate of skin tones of the top performing models.

Fig. 7, illustrates the mismatc rate on OULU-NPU and FPAD-g-23 datasets. Fig. 8 illustrates the mismatch rates (y-axis), reflecting the proportion of incorrect predictions for skin tone categories (x-axis) such as Brown (B), Light (L),Tan (T), Intermediate (I), Dark (D), and Very Light (VL). Both the figures suggests that baselines struggle to generalize across a diverse range of skin tones, leading to unequal performance on various skintones. However, ColorCubeNet substantially reduces this disparity, achieving more than a 50% reduction in mismatch rates across all skin tones when compared to traditional RGB-based models.

## A.5 ABLATION STUDY RESULTS

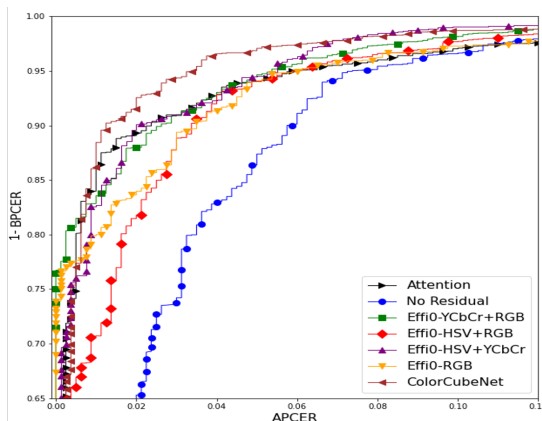

Figure 9: Receiver Operating Characteristic curve illustrating the performance of different ablation study configurations.

| Backbone | | | Channel | Concatnation | ACC% | EER% |
|---|---|---|---|---|---|---|
| RGB | HSV | YCbCr | Attention | | | |
| ✓ | x | x | ✓ | ✓ | 96.52 | 5.33 |
| ✓ | ✓ | x | ✓ | ✓ | 96.41 | 5.63 |
| x | ✓ | ✓ | ✓ | ✓ | 96.10 | 5.26 |
| ✓ | ✓ | ✓ | x | ✓ | 94.89 | 6.60 |
| ✓ | ✓ | ✓ | ✓ | x | 95.42 | 5.45 |
| ✓ | ✓ | ✓ | ✓ | ✓ | **97.34** | **4.41** |

Table 4: Ablation study using FPAD-G-23, This table shows Equal Error Rate% and Accuracy%

Impact of Channel Attention on various datasets.

|  | Attention | | |
|---|---|---|---|
| Datasets | Self | Channel | Spatial |
| Celeb-A | 4.68 | 4.15 | 4.84 |
| OULU NPU | 4.35 | 3.49 | 4.57 |
| Synth-Spoof | 0 | 0 | 0 |
| FPAD-i-22 | 3.29 | 2.29 | 3.27 |
| FPAD-g-23 | 6.43 | 3.96 | 4.9 |
| IIIT-D | 2.39 | 0.78 | 1.25 |

Table 5: Table shows EER% of Self, Channel and spatial attetnion mechanisms on ColorCubeNet

The channel-attention blocks focus on the most informative features within each color channel. It first applies global average pooling and global max pooling to the feature map, reducing each channel to two distinct values representing the average and maximum activations across the spatial dimensions.

## A.6 GRADCAM

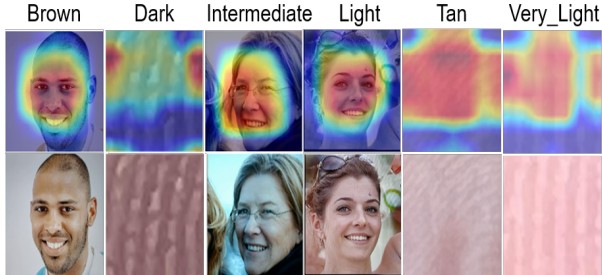

Figure 10: Grad-Cam samples

Grad-CAM, or Gradient-weighted Class Activation Mapping, comes up with a "class activation map" showing those important image regions contributing most to a target prediction. Let's compute the weights $\alpha_k^c$ for neuron $k$ and target class $c$: the global average of the gradients coming back from the output unit belonging to class $c$ onto the featre maps $A$ of convolution layers.

$$\alpha_k^c = \frac{1}{Z} \sum_i \sum_j \frac{\partial y^c}{\partial A_{ij}^k} \tag{9}$$

Where $\frac{\partial y^c}{\partial A_{ij}^k}$ is the gradient of the score for class $c$, $y^c$, concerning the feature map $A^k$ at spatial location $(i, j)$, and $Z$ is the total number of pixels in the feature map. This process helps identify the parts of the image that are most influential in making the classification decision, giving a clear visual interpretation of how this model's focus may vary across skin tones. In the model we used final residual layer after batch normalization for Grad-Cam.

