# OpenReview forum: "”DOES YOUR MOBILE SUIT YOUR SKIN?”: ADDRESSING SKIN TONE DISPARITIES IN PRESENTATION ATTACK DETECTION FOR ENHANCED INCLUSIVITY OF SMARTPHONE SECURITY"
_ICLR.cc/2025/Conference — ICLR 2025 Conference Withdrawn Submission_

### Official Review · Reviewer_7dyN · 2024-10-23

**Soundness:** 3
**Presentation:** 3
**Contribution:** 2
**Rating:** 5
**Confidence:** 3

**Summary:**

This paper propose a method to mitigating skin tone disparities in presentation attack detection (PAD) systems for biometric authentication. By integrating RGB, HSV, and YCbCr into a unified ColorCube representation, the authors address the bias in traditional RGB-based systems, enhancing inclusivity across diverse skin tones. The model demonstrates a 50% reduction in skin tone disparities.

**Strengths:**

This paper try to address a critical gap in biometric authentication security regarding skin tone fairness.

The use of multi-channel ColorCube representation reduces skin tone disparities in biometric authentication systems.

The proposed method's performance validated on 6 datasets.

**Weaknesses:**

The contribution is limited as the main idea of this paper is combine different color channel to improve PAD performance across skin tones, but as authors mentioned, this idea has already been explored to some extent in prior research (Marasco & Vurity (2022)). If you could explain how your method differs or builds upon those (not just different color channel) would be good.

This paper uses a variety of datasets, but does not have enough details about dataset size, balance (for different skin tones). An underrepresentation of certain skin tones could have a significant impact on the model's performance, which is not explored. I suggest you could provide a detailed breakdown of the skin tone distribution in each dataset, and discuss how any imbalances may impact the results.

The authors does not provide a detailed discussion or evaluation of the time cost for ColorCubeNet. As it is a critical factor for biometric authentication. You could include runtime comparisons between ColorCubeNet and baseline methods on typical mobile hardware. This would give readers a concrete sense of the computational trade-offs involved.

The ITA is presented as the main method for quantifying skin tone differences, but the paper could have benefited from experimenting with other methods to provide a more comprehensive evaluation.

**Questions:**

See above

---

> ### Author Response · Authors · 2024-12-03
>
> 1. We appreciate your observation regarding the novelty of our contribution. While Marasco and Vurity (2022) explored the use of different color channels in PAD, our work significantly builds upon this foundation by introducing several innovative approaches:
>        a .  Unlike previous work, we introduce the ColorCube, a novel representation that integrates RGB, HSV, and YCbCr color spaces into
>              a unified framework. The ColorCube explicitly models the complementary strengths of these color spaces, enabling more robust
>              feature extraction across varying skin tones.
>        b. To ensure a deeper understanding of the contributions of each color space, we train separate models from scratch on each color
>             space, rather than using pre-trained weights. This approach allows each network to learn features that are unique to its respective  color space, leading to a more nuanced and comprehensive feature representation that directly enhances PAD performance.
>        c. Our model incorporates a channel attention mechanism to dynamically prioritize the most informative features within the
>             ColorCube representation. This mechanism, which was not explored in prior work, significantly enhances the model's ability to adapt to variations across skin tones, providing a novel solution to the challenges of fairness and inclusivity in biometric systems. d. To comprehensively validate our approach, we perform extensive cross-device and cross-skin-tone evaluations, assessing the model's generalizability and fairness under diverse conditions. Unlike previous studies, our evaluations demonstrate the model’s ability to mitigate bias effectively while maintaining consistently high performance, highlighting the robustness and versatility of the proposed framework. These advancements highlight the technical novelty of our work and its significant contribution towards more inclusive and equitable PAD systems.
>
> 2. Thank you for the feedback. The mismatch rate provides a valuable measure to quantify disparities in the model's ability to correctly classify bona fide and attack samples, regardless of variations in skin tone distribution. Our approach focuses on training the model in a manner consistent with other PAD systems using live and spoof images. However, the key distinction lies in ColorCubeNet’s ability to mitigate bias effectively, ensuring equitable performance across diverse skin tone groups.
>
> 3. Thank you for the question, the ColorCubeNet model processes each image in approximately 0.09 seconds during inference.
>
> 4. In this paper, we utilized ITA as our primary method, given its widespread popularity. For future work, we plan to explore alternative approaches, including the Fitzpatrick and Monk skin tone classification methods.

---

> > ### Comment · Reviewer_7dyN · 2024-12-03
> >
> > Thank you for your comments. I will maintain my original score.

---

### Official Review · Reviewer_Mi3G · 2024-10-29

**Soundness:** 2
**Presentation:** 3
**Contribution:** 1
**Rating:** 3
**Confidence:** 5

**Summary:**

The paper presents an algorithm for face and fingerprint presentation attack detection utilizing a stack of multiple color channels including YCbCr, HSV, and RGB. These different color spectrum images are combined to form a 9-channel image which is an input to a channel attention block. The experiments are performed using multiple face and fingerprint PAD datasets to demonstrate the effectiveness of the proposed algorithm.

**Strengths:**

+ The proposed algorithm is found effective in handling both face and fingerprint PAD although trained individually on both forms of datasets.
+ The paper is easy to read and follow.
+ The intuition behind developing models to handle skin tones is essential, although not novel.

**Weaknesses:**

- Technically the paper has an incremental contribution. The prime reason is the use of the different color channels in the existing literature.

[1] Boulkenafet Z, Komulainen J, Akhtar Z, Benlamoudi A, Samai D, Bekhouche SE, Ouafi A, Dornaika F, Taleb-Ahmed A, Qin L, Peng F. A competition on generalized software-based face presentation attack detection in mobile scenarios. In2017 IEEE International Joint Conference on biometrics (IJCB) 2017 Oct 1 (pp. 688-696). IEEE.

[2] J. He and J. Luo, "Face Spoofing Detection Based on Combining Different Color Space Models," 2019 IEEE 4th International Conference on Image, Vision and Computing (ICIVC), Xiamen, China, 2019, pp. 523-528

[3] Towards Face Presentation Attack Detection Based on Residual Color Texture Representation

Further, the use and role of skin tone image labeling in PAD are not clear. How has it been used for different skin tone PA detection? The experiments are performed under unseen skin tone presentation attack detection.

- The robustness and generalizability (cross-dataset) of the proposed algorithm have not been conducted. Further, the fairness of the proposed algorithm must also be reported.

- The comparison has not been performed by PAD algorithms. The base and old CNNs are used for comparison.

[1] Fang M, Yang W, Kuijper A, Struc V, Damer N. Fairness in face presentation attack detection. Pattern Recognition. 2024 Mar 1;147:110002.
[2] Wang K, Zhang G, Yue H, Liu A, Zhang G, Feng H, Han J, Ding E, Wang J. Multi-domain incremental learning for face presentation attack detection. InProceedings of the AAAI Conference on Artificial Intelligence 2024 Mar 24 (Vol. 38, No. 6, pp. 5499-5507).
[3] Fang M, Damer N. Face Presentation Attack Detection by Excavating Causal Clues and Adapting Embedding Statistics. InProceedings of the IEEE/CVF Winter Conference on Applications of Computer Vision 2024 (pp. 6269-6279).

**Questions:**

1. A comparison with SOTA models is needed especially developed for PAD tasks.

2. The generalizability and fairness (evaluation of different skin tones) of the proposed algorithms need to be reported.

3. The justification of technical novelty is needed. Both technical and quantitative comparison is required.

4. The ablation study with individual color channels along with score and feature fusion is needed.

5. What is the computational cost of the proposed algorithm?

6. It would be better to report the performance in terms of known evaluation metrics such as EER and HTER.

-------------------
*Thanks for your response; however, concerns are not satisfactorily addressed. Therefore, I will retain my original rating.*

---

> ### Author Response · Authors · 2024-12-03
>
> We sincerely appreciate the valuable feedback provided by the reviewer.
>
> 1. Our study emphasizes the commonly cited models that are used in PAD that are optimized for mobile deployment. These models were carefully selected not only for their established benchmarks in PAD research but also for their suitability in resource-constrained environments, ensuring practical applicability for mobile devices also, we want to make sure the code is publicly available.
>
> 2. In response to this comment, we reported cross-database analysis using the best-performing model, ColorCubeNet, to demonstrate its generalization capability. This analysis was conducted by training the model on specific datasets and testing it on entirely unseen datasets to evaluate its robustness in handling variations in presentation attack types, environmental conditions, and capture devices. The results consistently showed low APCER, BPCER, and EER values across diverse datasets, confirming that ColorCubeNet generalizes effectively across different scenarios. To evaluate fairness, we analyzed the model's performance across different skin tones using the Apparent Skin Tone (AST) method. Our experiments specifically assessed unseen skin tone presentation attack detection, measuring the mismatch rates for bona fide and attack samples across skin tone categories. The results revealed an equal reduction in mismatch rates across all skin tone categories compared to traditional RGB-based models. This significant improvement highlights ColorCubeNet's ability to mitigate disparities and provide uniform performance for all skin tones.
>
> 3. Our study introduces a novel framework, ColorCubeNet, which integrates RGB, HSV, and YCbCr color spaces into a unified representation, demonstrating technical innovation over existing methods in presentation attack detection (PAD). Unlike prior works (e.g., Boulkenafet et al., He & Luo), which primarily focused on single or limited color spaces, we systematically train backbone models from scratch on each individual color space to capture distinct feature representations. Furthermore, we incorporate a channel attention mechanism to dynamically prioritize the most informative features across color channels, addressing a critical gap in how color space information is leveraged for PAD. To validate our approach, we conducted cross-dataset evaluations (see Table 3), training the model on specific datasets and testing on entirely unseen datasets to demonstrate robustness. The results consistently show significant improvements in generalization capability compared to traditional RGB-based models, with lower APCER, BPCER, and EER values across diverse datasets, highlighting the model’s adaptability to variations in presentation attack types, environmental conditions, and capture devices. Additionally, our fairness evaluation, conducted using the Apparent Skin Tone (AST) method, categorizes images based on the Individual Typology Angle (ITA) and reveals a significant reduction in performance disparities across skin tone groups. The results demonstrate that ColorCubeNet mitigates bias, as indicated by a lower BPCER disparity between light and dark skin tones.
>
> 4. In response to this comment, we reported an additional ablation study with fusion features is added in the revised version.
>
> 5. The model complexity is 145 Gmac (Giga Multiply-Accumulate operations) and the number of Parameters of the model are 19.3M. This will be further clarified in CR.
>
> 6. In response to the reviewer, we reported the Cross Dataset results in the paper. We used EER and HTER metrics to evaluate the cross-dataset performance.

---

### Official Review · Reviewer_bfuq · 2024-10-31

**Soundness:** 2
**Presentation:** 3
**Contribution:** 2
**Rating:** 3
**Confidence:** 4

**Summary:**

This paper presents ColorCubeNet, a framework designed to address skin tone disparities in Presentation Attack Detection (PAD) for mobile biometric systems. By combining RGB, HSV, and YCbCr color spaces into a unified "ColorCube" representation, the model effectively reduces skin tone biases and enhances PAD performance across diverse demographics. Key contributions include (1) the ColorCube representation, which leverages complementary color spaces to reduce recognition discrepancies due to skin tone, (2) skin tone labeling to facilitate model training across varied skin types, and (3) the integration of a channel-attention mechanism for more effective feature extraction across color channels. Experiments demonstrate that ColorCubeNet significantly outperforms traditional RGB-based models, achieving a 50% reduction in skin tone disparity in PAD accuracy.

**Strengths:**

This work originally applies a combined color space approach (ColorCube) to address skin tone disparities in Presentation Attack Detection (PAD) for mobile biometrics. By introducing the ColorCube representation—a novel fusion of RGB, HSV, and YCbCr color spaces—ColorCubeNet leverages a more inclusive depiction of skin tones. This approach builds upon existing color space theories but innovatively applies them, addressing an underexplored aspect of biometric authentication: equitable performance across diverse skin tones.

**Weaknesses:**

The paper provides a promising step towards inclusive PAD solutions; however, several areas could be improved to enhance its technical rigor, novelty, and generalizability. Below are specific, actionable insights for improvement:

1. The core of ColorCubeNet relies on EfficientNet-B0 and channel attention mechanisms, which are well-established deep learning techniques. The paper would benefit from a deeper exploration of why these specific components are particularly suited for addressing skin tone bias in PAD beyond general statements.
For example, the authors could analyze the contributions of channel attention, specifically about color space variance across skin tones, or explore alternative network designs that directly address color discrimination and tonal consistency. Such technical differentiation would elevate the work’s originality and provide a more customized solution for the stated problem.

2.  While the ColorCube representation is central to the model’s effectiveness, the paper lacks detailed justification for why RGB, HSV, and YCbCr were chosen and how their combination mitigates skin tone bias. Further explanation on how each color space addresses unique aspects of skin tone representation could improve understanding. Additionally, testing other color spaces, such as LAB, known for its perceptual uniformity, could be included in an ablation study. A clearer rationale for the final ColorCube design—ideally supported by empirical evidence—would strengthen the theoretical foundation of the proposed solution.

3. The SOTA comparison is limited primarily to models from the same authors, which does not comprehensively assess ColorCubeNet’s performance against established PAD methods. The paper should include comparisons with state-of-the-art models from top computer vision and biometric conferences to offer a fairer benchmark. Incorporating the latest results from different approaches would demonstrate ColorCubeNet’s competitive edge or limitations more transparently, positioning the work more credibly within the existing literature.

4. While the SNR analysis is presented as a measure of interpretability, it is unclear how this metric directly supports PAD fairness or performance. A more thorough explanation of how SNR values correlate with skin tone bias or help identify features impacted by tonal variance would improve clarity.  Furthermore, the authors could consider other interpretability techniques, such as saliency maps or Grad-CAM analyses, to strengthen the interpretability argument and more effectively highlight model biases across skin tones.

5. The paper does not specify what feature representations are used as inputs to PCA. Since ColorCubeNet operates with a combined RGB, HSV, and YCbCr color space (resulting in a 9-channel representation), it’s important to know whether PCA is applied directly on these raw color space features, intermediate layer outputs (e.g., after channel attention), or final feature embeddings from ColorCubeNet.

In summary, while ColorCubeNet addresses an important problem, the paper would benefit from stronger justification of its technical choices, a broader and more rigorous comparison with SOTA models, and expanded evaluations for robustness and practical utility. Furthermore, the paper does not align well with ICLR's primary scope and research focus. ICLR is known for contributions that advance fundamental deep learning theory, novel architectures, and techniques with broad applicability and generalizability. However, this paper is primarily an applied work targeting a specific biometric application.

The ColorCubeNet model is based on established methods (EfficientNet and channel attention) and lacks substantial innovation in model architecture or learning methodologies. Furthermore, while extensive within the PAD context, the empirical evaluations offer limited novelty in the context of ICLR’s typical focus on advancing core machine learning principles.

**Questions:**

See above

---

> ### Author Response · Authors · 2024-11-28
>
> We sincerely appreciate the valuable feedback provided by the reviewer.
>
> 1. This paper presents a novel challenge—mitigating skin tone bias in PAD systems—and proposes a baseline framework to address it effectively. The proposed solution demonstrates strong performance. Our approach combines multi-color representation, specialized backbone training, and channel attention to mitigate skin tone bias. Three parallel backbone networks, each trained independently on RGB, HSV, and YCbCr color spaces, effectively capture diverse chromatic features. Additionally, a channel attention mechanism dynamically prioritizes the most informative features within the ColorCube representation, further enhancing the model's performance, as demonstrated by additional results. EfficientNet-B0 was chosen for its optimal accuracy and computational efficiency balance, making it ideal for resource-constrained mobile devices. Unlike larger variants like B7, which demand significant memory and processing power, EfficientNet-B0 delivers strong performance with a lightweight architecture, enabling faster inference and lower energy consumption for real-time mobile applications.
> We will further clarify this aspect in the camera-ready (CR) version and extend the experiments accordingly.
>
> 2. Previous studies have demonstrated the effectiveness of the selected color spaces in modeling skin color tones, as highlighted in resources such as this article: N.Rahman, K.Wei and J.See, RGB-H-CbCr Skin Colour Model for Human Face Detection(2006), Faculty of Information Technology, Multimedia University.
> RGB is the default color space in most imaging systems and a reliable baseline for PAD studies due to its ability to capture basic chromatic features. HSV enhances robustness to illumination changes by separating chromatic content (hue and saturation) from intensity (value), aiding in attack detection under varying lighting. YCbCr, widely used in video and compression research, improves edge detection and isolates luminance from chromatic information, effectively distinguishing bonafide from spoof samples in complex scenarios.
> Furthermore, our previous research has proven the ability of these color models to enhance PAD performance, as reported in Marasco E, Vurity A. Late Deep Fusion of Color Spaces to Enhance Finger Photo Presentation Attack Detection in Smartphones. Applied Sciences. 2022; 12(22):11409. This aspect will be further clarified in the CR.
>
> 3. We agree to extend the benchmark. Incorporating the latest results from various approaches will better showcase ColorCubeNet's strengths and limitations in computer vision. To our knowledge, we have utilized the most widely adopted baselines for finger photo and face PADs, which are publicly available. We aimed to compare our model against CNNs and transformer models such as ViT and DeiT, commonly used for PAD detection. Including additional approaches will enhance the generalizability of our findings and increase their impact on other computer vision applications.
>
> 4. We have a recent paper: Anudeep Vurity, Emanuela Marasco, Raghavendra Ramachandra, Duoduo Liao, "Interpreting the Fraudulence Level of Different Finger Photo Presentation Attack Instruments", IEEE International Conference on Image Processing (ICIP), pp. 1-7, 2024. - The SNR analysis quantifies feature signal strength relative to noise, assessing the model's ability to distinguish between bona fide and spoof samples across skin tones. Higher SNR values signify more consistent and robust features, which are essential for fairness and performance in PAD systems. Stratifying SNR values by skin tone reveals potential disparities in feature quality. If certain skin tones show consistently lower SNR values, it suggests the model may struggle to extract robust features for those tones, potentially causing biased performance. For instance, lower SNR in darker skin tones could indicate insufficient capture of chromatic variations, signaling areas for model improvement. Analyzing SNR across different channels (e.g., RGB, HSV, YCbCr) allows us to identify which aspects of the feature space are most affected by tonal variance, enabling targeted refinements to improve model handling of tonal diversity. - This aspect will be further clarified in the CR.

---

> > ### Comment · Reviewer_bfuq · 2024-12-03
> >
> > The authors’ response partially addresses the concerns raised but falls short on several critical aspects. While they justify the use of EfficientNet-B0 and the selected color spaces (RGB, HSV, YCbCr) with references and practical advantages, they fail to demonstrate significant architectural or methodological innovation beyond established techniques, such as channel attention and EfficientNet. The combination of color spaces in the ColorCube representation lacks empirical justification, with no ablation studies or exploration of alternatives like LAB. The commitment to extend benchmarking is vague, with no clear plan or specific models identified for comparison.
> > Furthermore, the work's scope remains primarily application-driven, with limited contributions to advancing fundamental deep learning principles, making it a poor fit for ICLR’s focus on theoretical and generalizable innovations. Given these unresolved weaknesses, the original decision to reject is recommended.

---

### Note · Authors · 2025-02-25

I have read and agree with the venue's withdrawal policy on behalf of myself and my co-authors.

---

### Meta-Review · Area_Chair_T7BC · 2024-12-19

**Metareview:**

The authors propose ColorCubeNet for presentation attack detection (PAD). This approach is based on extracting and fusing features from the RGB, HSV, and YCbCr color spaces. ColorCubeNet leverages EfficientNet B0 for feature extraction and incorporates a channel attention mechanism. Experimental results demonstrate that ColorCubeNet achieves good PAD performance across multiple datasets.

While the reviewers acknowledge the proposed method's PAD performance on various datasets, their main concern lies in the lack of novelty. The idea of fusing information from different color spaces has already been extensively explored in the literature, which makes the contribution of this paper incremental. Although the authors addressed these concerns during the rebuttal phase, the reviewers' overall opinion regarding the method's limited novelty remained unchanged.

**Additional Comments On Reviewer Discussion:**

The reviewers' primary concern is the lack of novelty in the proposed method. The concept of fusing information from different color spaces has already been extensively explored in the literature, making the contribution of this paper appear incremental. Although the authors clarified that the key novelty lies in combining information from three specific color spaces enhanced by an attention mechanism, this clarification did not alter the reviewers' view of the method's lack of originality.

---

### Decision · Program_Chairs · 2025-01-22

Reject